# Generation of a Metal Ion Beam Using a Vacuum Magnetron Discharge

Alexey V. Vizir [1,*], Efim M. Oks [1,2], Maxim V. Shandrikov [1] and Georgy Yu. Yushkov [1]

1   Institute of High Current Electronics, Siberian Branch of the Russian Academy of Science, 634055 Tomsk, Russia; oks@fet.tusur.ru (E.M.O.); shandrikov@opee.hcei.tsc.ru (M.V.S.); gyushkov@mail.ru (G.Y.Y.)
2   Tomsk State University of Control Systems and Radioelectronics, 634050 Tomsk, Russia
*   Correspondence: vizir@opee.hcei.tsc.ru

**Abstract:** We have designed, fabricated and characterized an ion source based on a vacuum magnetron discharge. The magnetron discharge is initiated by a vacuum arc discharge, the plasma of which flows onto the magnetron sputtering target working surface. The vacuum arc material is usually the same as that of the magnetron target. The discharges operate at a residual pressure of $3 \times 10^{-6}$ Torr without working gas feed. Pulses of vacuum arc (30 μs) and magnetron discharge (up to 300 μs) are applied simultaneously. After ignition by the vacuum arc, the magnetron discharge runs in a self-sustained mode. Cu–Cu, Ag–Ag, Zn–Zn, and Pb–Pb pairs of magnetron target material and vacuum arc cathode material were tested, as well as mixed pairs; for example, Cu vacuum arc cathode and Pb magnetron target. An ion beam was extracted from the discharge plasma by applying an accelerating voltage of up to 20 kV between the plasma expander and grounded electrodes. The ion beam collector current reached 80 mA. The ion beam composition, analyzed by a time-of-flight spectrometer, shows that the beam consists mainly of singly-charged (about 90%) and doubly-charged (about 10% current fraction) magnetron target material ions. The ion beam radial current density non-uniformity was as low as $\pm 5\%$ over a diameter of 6.6 cm, which is the diameter of the source output aperture.

**Keywords:** magnetron sputtering; vacuum (gasless) magnetron; film deposition; plasma ion composition measurement

## 1. Introduction

The planar magnetron discharge [1,2], among all coating technologies [3] (pp. 77–139), [4] (pp. 30–30-10) over an extremely wide range of applications [4], is undoubtedly the most widely used due to the high rate of coating deposition, stable and sustainable parameters, and the ability to vary and control the physical, mechanical, electrical, chemical, optical, magnetic and other properties of deposited coatings. The use of a magnetron discharge plasma for ion beam production is limited by the significantly high operating pressure required for the discharge operation, usually $10^{-3}$ Torr and greater. Such high pressure is problematic for voltage hold-off in the ion beam accelerating gap, as well as for beam transport. Gasless sputtering has been proposed and studied [5,6], using a planar magnetron discharge initiated and supported by a supplementary vacuum arc discharge unit which is positioned off the magnetron axis. Disadvantages of this approach are that the vacuum arc plasma impacts the magnetron target surface asymmetrically, and the vacuum arc cathode is located relatively far from the magnetron target. Even though this kind of set-up has been investigated [5,6], it has not been used for ion beam formation, and the discharge plasma composition has not been studied. In prior work [7] we have developed and explored a planar gasless magnetron discharge initiated by a vacuum arc plasma gun located on the axis of a planar magnetron. The ion composition of the discharge

plasma was measured. Here we describe the parameters of an ion source based on a gasless magnetron discharge initiated by a vacuum arc unit positioned on the magnetron axis.

The implementation of a gasless magnetron will provide generation of a purely metallic plasma in the discharge. The use of such a plasma in an ion source will ensure the formation of metal ion beams. The demand for metal ions is associated with their use in accelerator technology [8], ion thrusters [9], as well as in technological installations of ion implantation for hardening the surface properties of various materials [10]. At present, ion sources based on a vacuum arc with a cathode spot are mainly used to generate beams of metal ions [11]. Compared to a vacuum arc, the principal advantage of the gasless magnetron system is the absence of a micro-droplet fraction. Another positive feature of a gasless magnetron is the fact that the ions of the metal plasma are mainly single charged. As a result, the generation of metal ion beams based on a gasless magnetron will ensure the absence of micro-drop impurities in the ion beam and make energy spread of such a beam less than a metal ion beam extracted from a vacuum arc ion source. Taking also into account higher stability and lower noising of the plasma parameters, the advantages of a gasless magnetron make ion sources based on it a real alternative to vacuum arc metal ion sources.

## 2. Materials and Methods

A schematic diagram of the ion source is shown in Figure 1. A planar magnetron discharge operates between target (Figure 1, point 1), with diameter 50 mm, and anode (2). A magnetic field in the magnetron discharge system is established by axially symmetric NdFeB permanent magnets (3) and a magnetic core (4) made of ferromagnetic steel. A vacuum arc discharge system (plasma gun) with cathode (5), anode (6) and initiating electrode (7) is installed on the axis of the magnetron discharge system. The vacuum arc plasma flow formed by the vacuum arc propagates toward the magnetron target. In conventional magnetrons [2], the discharge is ignited and operated in the working gas. In this work, the flow of the vacuum arc plasma onto the magnetron discharge target played the role of a medium in which the magnetron discharge was ignited and operated.

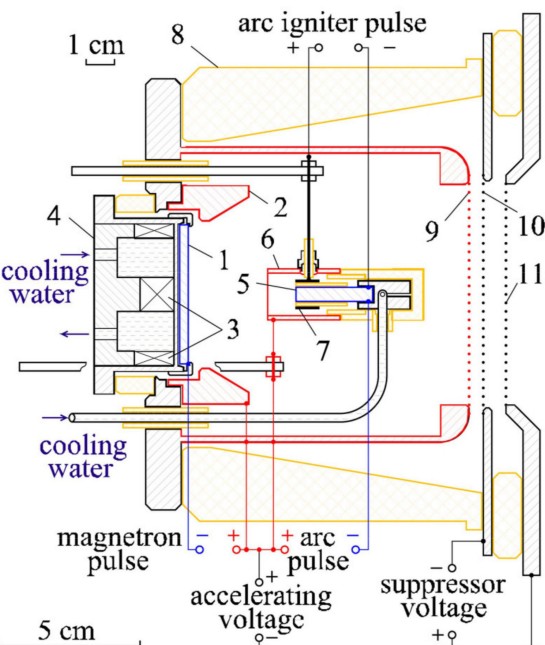

**Figure 1.** Schematic view of vacuum magnetron discharge ion source. 1—Magnetron target, 2—Magnetron anode, 3—Permanent magnets, 4—Magnetic circuit, 5—Vacuum arc cathode, 6—Vacuum arc anode, 7—Vacuum arc igniter electrode, 8—High voltage insulator, 9—Expander with emission grid, 10—Suppressor electrode, 11—Grounded electrode.

The anodes of the magnetron and vacuum arc are electrically connected and isolated from ground potential by high-voltage insulator (8). Water cooling of the cathode units of both the magnetron and the vacuum arc is supplied through insulating water decoupling with a channel length of 5 m and diameter 3 mm. The discharge plasma of a magnetron discharge, supported by the plasma of a vacuum arc, fills the expander (9), which is electrically connected to anodes 2 and 6 unless otherwise specified. The expander metal grid with mesh size of 0.7 mm × 0.7 mm forms the plasma emission surface. The expander, the grid and the discharge anodes are held at a steady high accelerating potential of up to +20 kV respect to ground potential. The emission grid of the expander (9), the suppressor grid (10) and the grounded grid (11) together form the ion beam extraction system.

A photograph of the ion source plasma generation system is shown in Figure 2a. Figure 2b shows the vacuum arc plasma generator as viewed from the magnetron surface (in the photo it is dismounted).

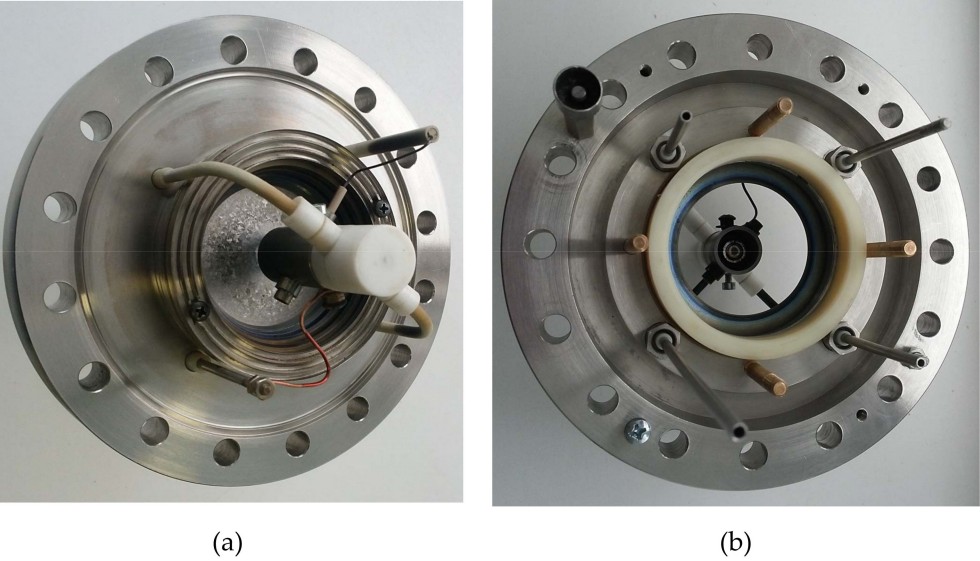

(a)           (b)

**Figure 2.** (**a**) External view of the ion source plasma generation system; (**b**) View of the vacuum arc plasma generator as seen from the magnetron surface.

The magnetron discharge is powered using an APEL-M-2HIPIMS-1500 pulsed power supply [12] with output voltage adjustable over the range 150–1500 V, pulse duration from 5 μs to 300 μs and current from 1 A to 100 A. The vacuum arc power supply employs a pulse forming network (PFN) to generate a current pulse with amplitude adjustable up to 200 A and duration 30 μs. A trigger pulse initiates the vacuum arc by flashover across a ceramic ring which was located between vacuum arc igniter electrode (Figure 1, point 7) and vacuum arc cathode (5). The voltage of trigger pulse was up to 10 kV and a pulse trigger current and the pulse duration were about of 15 A and 1 μs, respectively.

Power supplies of magnetron discharge, vacuum arc and trigger pulse are held at high positive accelerating voltage up to +20 kV. The pulse repetition rate is 2 pulses per second. It is possible to switch on all three power supplies at different times, but we have found that the magnetron and arc discharges are initiated most stably when all three power supplies are triggered simultaneously.

The mass-to-charge composition of the ion beam is analyzed by a time-of-flight spectrometer described in full elsewhere [13,14], utilizing a gate made in the form of a set of five pairs of concentric metal rings (plates) located in a plane perpendicular to the beam direction at a distance of 0.4 m from the ion source. The principle of operation of spectrometer gate is similar to the Bradbury–Nielsen gate [15]. For each pair of rings, the outer ring is grounded and the inner ring is connected to electronics that provide an ion-deflecting voltage pulse of amplitude of up to 2 kV and duration of 150 ns. The central

part of the gate is flat metal plate of diameter of 5 cm that blocks the direct passage of ion beam to ion beam detector when ion-deflecting voltage is not applied. The application of an ion-deflecting voltage pulse to the gate ensures that the ion beam is focused on the ion detector during the duration of this pulse. The ion detector is located on the beam axis at a distance of 1.2 m from the gate, and it was a secondary electron multiplier (SEM) VEU-1A operating in analog mode. The output current of the SEM anode is recorded by an oscilloscope, and beam ions with different charge to ion mass ratios on the oscillogram are formed peaks (see, for example, Figure 4 further in the Section 3) from which the composition of the ion beam was analyzed. Synchronization of the moment of application of the gate pulse relative to the moment of application of magnetron discharge and vacuum arc pulses was carried out from a six-channel generator of triggering pulses GI-1.

A titanium ion-beam collector plate 10 cm in diameter was located 25 cm from the ion source. When measuring the ion composition of the beam, the collector was turned (though a movable vacuum feed-through) so that its surface plane is parallel to the beam propagation direction. The radial distribution of ion beam current density was measured by a linear set of flat probes with common flat guard electrode. The ion beam collected surface area of each flat probe was $0.5 \times 0.5$ cm and distance between nearby probes was 1 cm.

The cylindrical vacuum chamber, with diameter 0.3 m and length 2 m, is pumped by a Varian TV 1001 Navigator turbomolecular pump with pumping speed 900 $l$/s. No working gas inlet for magnetron discharge was used. The base pressure in the vacuum chamber and in the discharge system is $3 \times 10^{-6}$ Torr.

## 3. Results

Typical magnetron discharge voltage and current and vacuum arc current pulses are shown in Figure 3; here, a Pb magnetron target and Pb vacuum arc cathode were used. We have also used Cu, Ag, and Zn as magnetron target and vacuum arc cathode materials [7]. One can see from Figure 3 that the self-sustained vacuum magnetron discharge voltage is about 600 V for a Pb target. At the beginning of the pulse, during vacuum arc ignition, the voltage drops to about 300 V due to support of the magnetron discharge by the vacuum arc plasma.

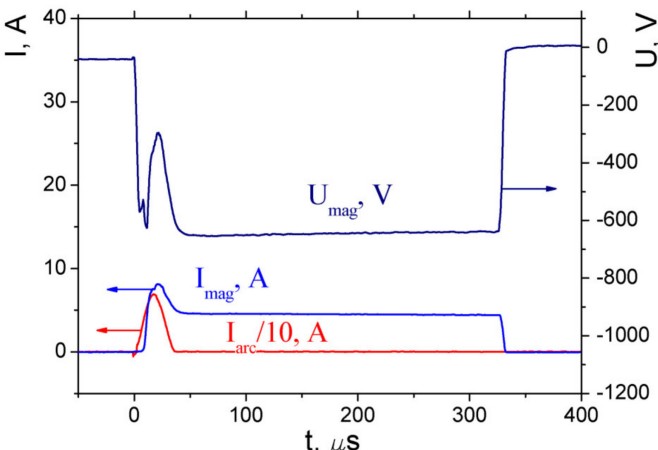

**Figure 3.** Magnetron discharge voltage and current, vacuum arc current. Pb magnetron target, Pb arc cathode.

With no vacuum arc, application of voltage between the magnetron cathode and anode does not lead to magnetron discharge ignition. A specific vacuum arc current threshold is needed for the magnetron discharge to be ignited. For instance, using a copper arc cathode and a copper magnetron target, a 50 A, 30 µs vacuum arc pulse leads to ignition of a Table 50 A, 300 µs magnetron discharge.

A time-of-flight spectrum of the ion beam formed using a Pb magnetron target and Pb vacuum arc cathode is shown in Figure 4. For approximately 30–40 µs after discharge

current pulse initiation, the beam contains ions of residual gas (hydrogen, nitrogen and oxygen of various charge states) in addition to ions of the cathode material. After this 30–40 µs period, the fraction of gaseous ions decreases from several tens of percent to less than 1%. This thus implies that the magnetron discharge takes place exclusively in the vapor of the magnetron target material after the end of the vacuum arc current pulse.

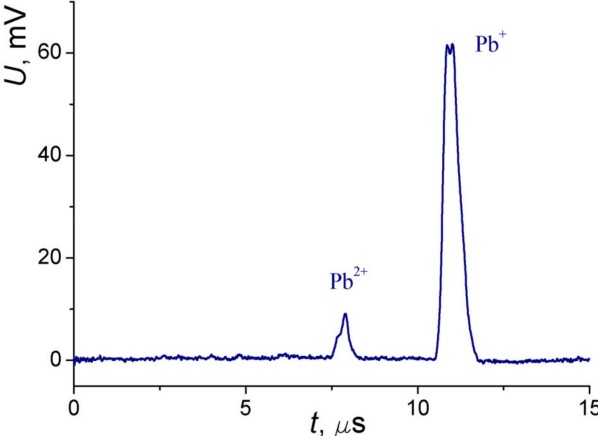

**Figure 4.** Time-of-flight ion spectrum of beam formed with a Pb magnetron target and a Pb vacuum arc cathode. Magnetron current is 5 A, arc current is 100 A, time delay after beginning of discharges is 100 µs.

Note that when using different vacuum arc and magnetron metals, for example copper vacuum arc cathode and lead magnetron target, the fraction of vacuum arc material (e.g., copper ions) decreases to less than 1% by the end of the vacuum arc current pulse. This may be because, despite the fact that the arc current is significantly greater than the magnetron current, the contribution of magnetron sputtering to the plasma composition exceeds the contribution from the vacuum arc plasma gun because of the higher magnetron discharge voltage. Thus, a persistent copper film does not form on the magnetron target surface.

The mean ion charge states of the magnetron target materials studied, (Cu, Ag, Zn, Pb) are much lower than those for a vacuum arc plasma [16]. For example, a mean ion charge state for Pb of 1.64 has been reported [16], but for the vacuum magnetron plasma studied in the present work the mean charge state is 1.13. This may be related to a lower discharge current for the magnetron and thus lower plasma density. The current-voltage characteristics of the ion source for a copper magnetron target and copper vacuum arc cathode (Figure 5) show that the ion beam current is up to 0.5% of the magnetron discharge current.

The total ion emission current from the magnetron plasma was measured (current to the expander 9 (Figure 1), plus current to the emission grid when the expander and grid are biased negative with respect to grounded anodes 2 and 6). The current had a distinct saturation at an expander bias of 50 V and was 0.34, 0.7 and 1.25 A for magnetron discharge currents of 6, 10 and 17 A, respectively (5.7%, 6.9% and 7.4% of the magnetron current, respectively) at zero vacuum arc discharge current (the ion current from the magnetron discharge plasma was measured 150 µs after the beginning of the magnetron discharge current pulse; the vacuum arc was off at this time) for a copper magnetron target and vacuum arc. The current in the collector circuit was several times less than the current in the accelerating voltage circuit. We note that the application of a negative voltage to the suppressor electrode (Figure 1, point 10) leads to a decrease in collector current of about 20%, implying that the suppressor indeed impedes the back-streaming secondary electron flux from the downstream beam transport region. However, we used a grid (wire mesh) system and not a multi-aperture beam acceleration system, and thus the electron flux emitted by the grid electrode (10) is not suppressed, in turn leading to increased emission current with increasing accelerating voltage (see Figure 5). Clearly it is

prudent to use a multi-aperture ion-optical system for the formation of ion beams at higher accelerating voltages.

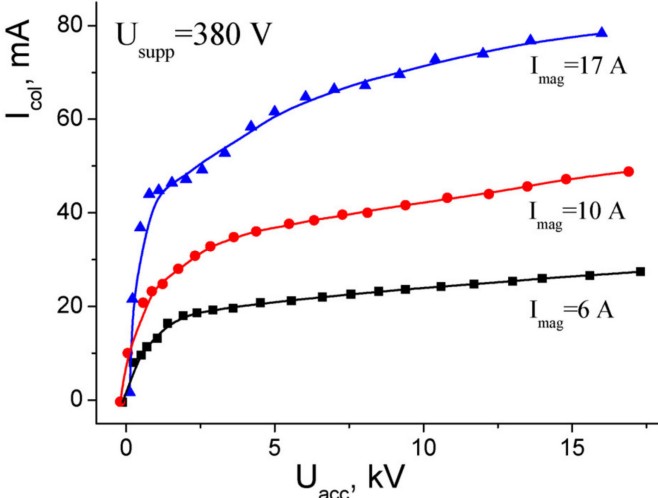

**Figure 5.** Ion beam current dependence on accelerating voltage for different magnetron discharge currents. Copper magnetron target, copper vacuum arc cathode.

To study how the uniformity of the ion beam changes at different distances from the ion source, the radial distributions of the ion beam current density were measured. The radial distribution of ion beam current density (Figure 6a) has a minimum on the system axis for downstream distances from the ion source less than 30 cm, evidently related to the central positioning of the vacuum arc plasma gun. At a downstream distance of 55 cm the distribution is more uniform, with no central minimum. At the same time, the beam diameter slightly increases.

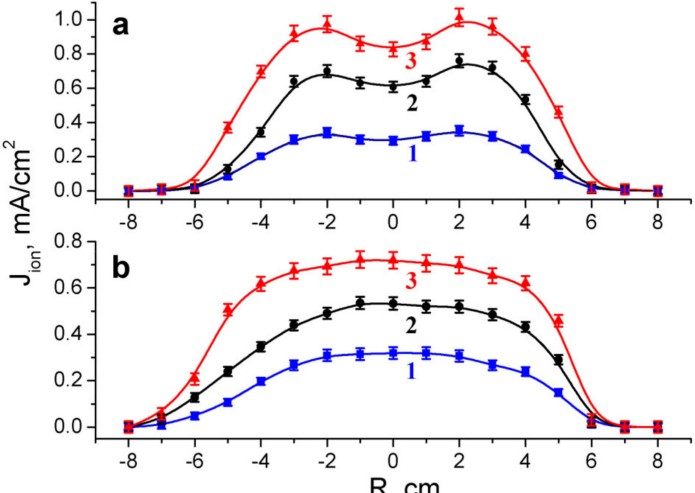

**Figure 6.** Radial distribution of ion beam current density. Distance between ion source and probe: (**a**) 30 cm, and (**b**) 55 cm. **1**—magnetron current 6 A, **2**—10 A, **3**—17 A. Copper magnetron target, Copper vacuum arc cathode. Ion source output diameter 6.6 cm.

For a magnetron discharge current of 17 A (Figure 6b, curve 3) the radial non-uniformity of the ion beam is ±5% over a diameter of 6.6 cm, which is the diameter of the ion source output aperture of the expander with emission grid.

## 4. Conclusions

We have developed and characterized a metal ion source based on a pulsed vacuum planar magnetron discharge operating without gas feed at a residual pressure of $3 \times 10^{-6}$ Torr and initiated by an on-axis vacuum arc plasma gun. The source forms ion beams of the magnetron target material, with a singly charged ion fraction about 90% and doubly charged ion fraction about 10%. At an accelerating voltage of 16 kV, the ion current reaches 80 mA with a magnetron discharge current of 17 A and vacuum arc current of 0 A. The radial non-uniformity of the beam current density is $\pm 5$% over a diameter of 6.6 cm, which is the diameter of the source output aperture.

The achieved parameters of the metal ion beam current are comparable to the ion current of a vacuum arc source. Higher efficiency of ion extraction from plasma, stability and low noise level of ion beam parameters, together with the absence of a droplet fraction, makes an ion source based on a gasless magnetron attractive for use in those areas where vacuum arc sources of metal ions dominate today. Further development of sources of metal ions based on a gasless magnetron will be associated with the generation of a wide range of ions of various metals and the study of the possibilities of increasing the current of the ion beam and its density while maintaining the high quality of the parameters of the ion beam.

**Author Contributions:** Conceptualization, A.V.V. and E.M.O.; methodology, G.Y.Y.; validation, G.Y.Y.; investigation, A.V.V. and M.V.S.; data curation, M.V.S.; writing—Original draft preparation, A.V.V.; writing—Review and editing, E.M.O.; supervision, E.M.O.; funding acquisition, E.M.O. All authors have read and agreed to the published version of the manuscript.

**Funding:** This research concerning the investigation of the discharge plasma and ion beam analysis was supported by the Russian Foundation for Basic Research (RFBR), grant number 19-48-700008, while the work concerning the ion source development was supported within the framework of the state assignment project FWRM-2021-0006.

**Acknowledgments:** The authors greatly appreciate Ian Brown (Berkeley Lab) for helpful discussion as well as for English correction.

**Conflicts of Interest:** The authors declare no conflict of interest. The funders had no role in the design of the study; in the collection, analyses, or interpretation of data; in the writing of the manuscript, or in the decision to publish the results.

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
