# Peer review of "Generation of a Metal Ion Beam Using a Vacuum Magnetron Discharge"

_plasma, doi:10.3390/plasma4020014_

Round 1

Reviewer 1 Report

The aim of the research is reasonably well explained. If the authors could add a couple of possible specific applications (medical physics, energetics?) for the development of such a type of wide source of metallic heavy ion beams, the work would be better justified and likely appeal to a wider audience.

I have a small number of corrections and suggestions.

(1) page 2, line 59: ….a metal “grid” with “mesh” size ….would be better.

(2) page 3, line 83: 2 pps, I am not sure about this abbreviation. “2 pulses per second” would seem better.

(3) page 3, lines 96-101: The meaning of this paragraph is unclear, notably the last sentence and should be rephrased. Time resolution is obtained by suitably delaying the 2kV, 150 ns pulse on the deflecting plates. This “extracted” ion pulse could then be mass analysed using e.g. a bending magnet spectrometer tuned to a particular value of (e/m), (2e/m), etc…With the TOF you can get them all at once. The authors must clarify their point here. Also, and perhaps that is the part of the explanations, they say nothing about how they synchronise the 2kV deflecting pulse with the magnetron/arc voltages and the type of delay line they use (a time delay of 100 microseconds is mentioned in the caption of Fig.4). They must give these details. 

(4) page 3, last paragraph. I cannot reconcile the “0.3 m diameter” of the vacuum chamber, therefore spherical? with its "length" of 2m?  “Turbomolecular” pump is better.

(5) page 4, line 132: “After” this 30- 40 ..period

(6) page 6, Figure 6. Typically Figures are read from top to bottom in alphabetical order (a), (b),… An error bar on the current density measurements is needed to show that the “dips” of Figure 6(a) are indeed physical. Also, what is the catchment surface area of each of the “flat probes”?

I recommend that the paper is published after the authors have satisfactorily addressed these points.

Author Response

Point-by-point our response to the reviewer’s comments are presented below.

The aim of the research is reasonably well explained. If the authors could add a couple of possible specific applications (medical physics, energetics?) for the development of such a type of wide source of metallic heavy ion beams, the work would be better justified and likely appeal to a wider audience.

The introduction of the article was completed by the following:

The implementation of a gasless magnetron will provide generation of a purely metallic plasma in the discharge. The use of such a plasma in an ion source will ensure the formation of metal ion beams. The demand for metal ions is associated with their use in accelerator technology [8], ion thrusters [9], as well as in technological installations of ion implantation for hardening the surface properties of various materials [10]. At present, ion sources based on a vacuum arc with a cathode spot are mainly used to generate beams of metal ions [11]. Compared to a vacuum arc, the principal advantage of the gasless magnetron system is the absence of a micro-droplet fraction. Another positive feature of a gasless magnetron is the fact that the ions of the metal plasma are mainly single charged. As a result, the generation of metal ion beams based on a gasless magnetron will ensure the absence of micro-drop impurities in the ion beam and make energy spread of such a beam less then metal ion beam extracted from a vacuum arc ion source. Taking also into account higher stability and lower noising of the plasma parameters, the advantages of a gasless magnetron make ion sources based on it a real alternative to vacuum arc metal ion sources.

I have a small number of corrections and suggestions.

(1) page 2, line 59: ….a metal “grid” with “mesh” size ….would be better.

 We agree with the comment of the Reviewer. The text has been changed as:

"The expander metal grid with mesh size of 0.7 x 0.7 mm forms the plasma emission surface."

(2) page 3, line 83: 2 pps, I am not sure about this abbreviation. “2 pulses per second” would seem better.

The abbreviation "pps" has been changed to "pulses per second".

(3) page 3, lines 96-101: The meaning of this paragraph is unclear, notably the last sentence and should be rephrased. Time resolution is obtained by suitably delaying the 2kV, 150 ns pulse on the deflecting plates. This “extracted” ion pulse could then be mass analysed using e.g. a bending magnet spectrometer tuned to a particular value of (e/m), (2e/m), etc…With the TOF you can get them all at once. The authors must clarify their point here. Also, and perhaps that is the part of the explanations, they say nothing about how they synchronise the 2kV deflecting pulse with the magnetron/arc voltages and the type of delay line they use (a time delay of 100 microseconds is mentioned in the caption of Fig.4). They must give these details.

The text was rewritten as following:

The mass-to-charge composition of the ion beam is analyzed by a time-of-flight spectrometer described in full elsewhere [13,14], utilizing a gate made in the form of a set of five pairs of concentric metal rings (plates) located in a plane perpendicular to the beam direction at a distance of 0.4 m from the ion source. The principle of operation of spectrometer gate is similar to the Bradbury-Nielson gate [15]. For each pair of rings, the outer ring is grounded and the inner ring is connected to electronics that provides an ion-deflecting voltage pulse of amplitude of up to 2 kV and duration of 150 ns. The central part of the gate is flat metal plate of diameter of 5 cm that blocks the direct passage of ion beam to ion beam detector when ion-deflecting voltage is not applied. The application of an ion-deflecting voltage pulse to the gate ensures that the ion beam is focused on the ion detector during the duration of this pulse. The ion detector is located on the beam axis at a distance of 1.2 m from the gate and it was a secondary electron multiplier (SEM) VEU-1A operating in analog mode. The output current of the SEM anode is recorded by an oscilloscope, and beam ions with different charge to ion mass ratios on the oscillogram are formed peaks (see, for example, figures 4 further in the Section 4) from which the composition of the ion beam was analyzed. Synchronization of the moment of application of the gate pulse relative to the moment of application of magnetron discharge and vacuum arc pulses was carried out from a six-channel generator of triggering pulses GI-1.

(4) page 3, last paragraph. I cannot reconcile the “0.3 m diameter” of the vacuum chamber, therefore spherical? with its "length" of 2m? “Turbomolecular” pump is better.

The following change of the text has been made:

"The cylindrical vacuum chamber, with diameter 0.3 m and length 2 m, is pumped by a Varian TV 1001 Navigator turbomolecular pump with pumping speed 900 l/s."

(5) page 4, line 132: “After” this 30- 40 ..period

The text has been changed as:

"After this 30 - 40 μs period..."

(6) page 6, Figure 6. Typically Figures are read from top to bottom in alphabetical order (a), (b),… An error bar on the current density measurements is needed to show that the “dips” of Figure 6(a) are indeed physical. Also, what is the catchment surface area of each of the “flat probes”? (AN)

The Figure 6 was changed and errors bars were added.

The collected surface area of probes was added to text as following:

"The ion beam collected surface area of each flat probe was 0.5 x 0.5 cm and distance between nearby probes was 1 cm."

I recommend that the paper is published after the authors have satisfactorily addressed these points.

We appreciated the work of the Reviewer in reading and evaluating our article and we want to thank him for this. We took into account all the comments of the Reviewer in the new version of the article and hope that the article is more understandable for our future readers.

Reviewer 2 Report

The work presented by the authors is brief and concise, but it lacks certain aspects that must be corrected to be published in this journal.

1) The authors comment on the importance of this type of discharge in technological applications but do not associate their results with such technological applications and the importance of these for improving the characteristics of the treated surfaces. This aspect is relevant and should be dealt with in this article to make it attractive to the scientific community. Also, the research carried out and the goodness of the results obtained would be enhanced.

2) The figures have a low resolution. They must be improved. Lack of uniformity.

3) There is only one bibliographic reference for the year 2020, the rest of the references are very old. Are there no more current references?

4) At the end of the introduction they should explain how the article is going to be developed.

5) The conclusions are concise and do not reveal the goodness of the results. No future jobs?

6) The materials and methods section should be improved, it's the wording, it is the most important section of the work but its explanation is not clear.

7) The uniformity of the discharge on the surface to be treated is not detailed. It is necessary to know if the deposition is done correctly.

8) In Figure 6, what is the reason for taking the distances of 30 cm and 55 cm, respectively? What are the characteristics of the probe used?

The authors must review and add the previous comments, in this way the work can be improved and considered for publication. A more real application of the measurements made with this experimental device is necessary.

Author Response

Our Point-by-point responses to the reviewer’s comments are presented below.

The work presented by the authors is brief and concise, but it lacks certain aspects that must be corrected to be published in this journal.

1) The authors comment on the importance of this type of discharge in technological applications but do not associate their results with such technological applications and the importance of these for improving the characteristics of the treated surfaces. This aspect is relevant and should be dealt with in this article to make it attractive to the scientific community. Also, the research carried out and the goodness of the results obtained would be enhanced.

The introduction of the article was completed by the following:

The implementation of a gasless magnetron will provide generation of a purely metallic plasma in the discharge. The use of such a plasma in an ion source will ensure the formation of metal ion beams. The demand for metal ions is associated with their use in accelerator technology [8], ion thrusters [9], as well as in technological installations of ion implantation for hardening the surface properties of various materials [10]. At present, ion sources based on a vacuum arc with a cathode spot are mainly used to generate beams of metal ions [11]. Compared to a vacuum arc, the principal advantage of the gasless magnetron system is the absence of a micro-droplet fraction. Another positive feature of a gasless magnetron is the fact that the ions of the metal plasma are mainly single charged. As a result, the generation of metal ion beams based on a gasless magnetron will ensure the absence of micro-drop impurities in the ion beam and make energy spread of such a beam less then metal ion beam extracted from a vacuum arc ion source. Taking also into account higher stability and lower noising of the plasma parameters, the advantages of a gasless magnetron make ion sources based on it a real alternative to vacuum arc metal ion sources.

2) The figures have a low resolution. They must be improved. Lack of uniformity.

All figures of article were improved.

3) There is only one bibliographic reference for the year 2020, the rest of the references are very old. Are there no more current references?

Additional references to literature sources have been added to the introduction [8-11], …..

4) At the end of the introduction they should explain how the article is going to be developed.

This was added in the conclusion

5) The conclusions are concise and do not reveal the goodness of the results. No future jobs?

The conclusion of the article was completed by the following:

The achieved parameters of the metal ion beam current are comparable to the ion current of a vacuum arc source. Higher efficiency of ion extraction from plasma, stability and low noise level of ion beam parameters, together with the absence of a droplet fraction, makes an ion source based on a gasless magnetron attractive for use in those areas where vacuum arc sources of metal ions dominate today. Further development of sources of metal ions based on a gasless magnetron will be associated with the generation of a wide range of ions of various metals and the study of the possibilities of increasing the current of the ion beam and its density while maintaining the high quality of the parameters of the ion beam.

6) The materials and methods section should be improved, it's the wording, it is the most important section of the work but its explanation is not clear.

The Section "Materials and methods" was rewritten. Please, see new version of the Section in resubmitted manuscript.

7) The uniformity of the discharge on the surface to be treated is not detailed. It is necessary to know if the deposition is done correctly.

In this work, we did not do and did not investigate the deposition of metal coatings by a magnetron discharge. The magnetron discharge was used only as a plasma generator from which the ion beam was extracted.

8) In Figure 6, what is the reason for taking the distances of 30 cm and 55 cm, respectively? What are the characteristics of the probe used?

For explanation why we presented radial distributions of the ion beam current density for distanses of 30 cm and 55 cm we added following:

"To study how the uniformity of the ion beam changes at different distances from the ion source, the radial distributions of the ion beam current density were measured."

The collected surface area of probes was defined to text of manuscript as following:

"The ion beam collected surface area of each flat probe was 0.5 x 0.5 cm and distance between nearby probes was 1 cm."

The authors must review and add the previous comments, in this way the work can be improved and considered for publication. A more real application of the measurements made with this experimental device is necessary.

We are deeply grateful to the Reviewer for his work in reading and evaluating the article. In the new version of the manuscript, we took into account all his comments and think that the article has become better and more understandable to readers.

Round 2

Reviewer 2 Report

The authors have taken into account the reviewer's indications and have reflected them in the text. Therefore, the work can be considered for publication in Plasma journal.